# Mechanisms of Microglia Proliferation in a Rat Model of Facial Nerve Anatomy

**DOI:** 10.3390/biology12081121

**Published:** 2023-08-11

**Authors:** Takashi Ishijima, Kazuyuki Nakajima

**Affiliations:** 1Graduate School of Science and Engineering, Soka University, Tokyo 192-8577, Japan; e21d5903@soka-u.jp; 2Glycan & Life Systems Integration Center, Soka University, Tokyo 192-8577, Japan

**Keywords:** microglia, proliferation, facial nerve, axotomy, M-CSF, cFms, cyclin, PCNA, p38, JNK

## Abstract

**Simple Summary:**

It has remained obscure whether microglia proliferate in the diseased or injured brain. This study analyzed the mitotic ability of microglia in vivo in a rat facial nerve axotomy model in which the blood cells do not enter the parenchyma. Microglia were found to proliferate in the axotomized facial nucleus at 5–7 days post-insult. They enhanced the levels of macrophage colony-stimulating factor (M-CSF) as a mitotic factor, its receptor cFms, and cell-cycle-related proteins, such as proliferating cell nuclear antigen (PCNA) and cyclins in vivo. In the M-CSF-dependent proliferation system in vitro, c-Jun N-terminal kinase (JNK) and p38 were shown to function in the induction of PCNA/cyclins and cFms in microglia, respectively. These analyses revealed that some specific signaling cascades are linked to the proliferative reaction of microglia. Microglia in the axotomized facial nucleus were considered to exhibit a mitotic ability to rescue injured motoneurons as neurotrophic cells.

**Abstract:**

Although microglia exist as a minor glial cell type in the normal state of the brain, they increase in number in response to various disorders and insults. However, it remains unclear whether microglia proliferate in the affected area, and the mechanism of the proliferation has long attracted the attention of researchers. We analyzed microglial mitosis using a facial nerve transection model in which the blood–brain barrier is left unimpaired when the nerves are axotomized. Our results showed that the levels of macrophage colony-stimulating factor (M-CSF), cFms (the receptor for M-CSF), cyclin A/D, and proliferating cell nuclear antigen (PCNA) were increased in microglia in the axotomized facial nucleus (axotFN). In vitro experiments revealed that M-CSF induced cFms, cyclin A/D, and PCNA in microglia, suggesting that microglia proliferate in response to M-CSF in vivo. In addition, M-CSF caused the activation of c-Jun N-terminal kinase (JNK) and p38, and the specific inhibitors of JNK and p38 arrested the microglial mitosis. JNK and p38 were shown to play roles in the induction of cyclins/PCNA and cFms, respectively. cFms was suggested to be induced through a signaling cascade of p38-mitogen- and stress-activated kinase-1 (MSK1)-cAMP-responsive element binding protein (CREB) and/or p38-activating transcription factor 2 (ATF2). Microglia proliferating in the axotFN are anticipated to serve as neuroprotective cells by supplying neurotrophic factors and/or scavenging excite toxins and reactive oxygen radicals.

## 1. Introduction

The central nervous system (CNS) is constructed from heterogenous cell types, including neurons and glial cells. Glial cells are grouped into macroglia (astrocytes and oligodendrocytes) and microglia [1,2]. In the early 1900s, the CNS was thought to be made up exclusively of neurons (the first element) and astrocytes (the second element), but in 1913, Cajal recognized that there was also a distinct “third element” of the CNS [3,4]. Subsequently, a pupil of his, del Rio-Hortega, selectively dyed microglia as part of his analysis of the third element [5]. For this reason, it is considered that microglia were discovered by Rio-Hortega. Like his teacher, Rio-Hortega believed that microglia originated from mesodermal cells. However, Rio-Hortega expanded this concept, suggesting that microglia originated from mesodermal precursor cells (extracerebral cells) that enter the brain and adopt an ameboid morphology, followed by a ramified morphology in the developing brain [5].

Soon after their discovery, however, microglia began to engender confusion in regard to their cell origin (cell lineage). In the 1980s, two groups put forward the idea that microglia were derived from monocytes, and this theory became widely popular [6,7]. According to this theory, monocytes produced in the bone marrow infiltrate the brain parenchyma from the bloodstream during the early developmental stage of the brain and then gradually transform into microglia in the parenchyma. This bone-marrow-derived monocytes theory was readily accepted by many researchers worldwide. However, subsequent analyses of radiation bone marrow chimeras left the theory open to question [8,9], since they revealed that bone-marrow-derived monocytes do not usually transform to ramified microglia in the adult parenchyma.

At the same time, some research groups advocated that microglia originate from neuroectodermal cells. One group described that glioblasts differentiate into astrocytes, oligodendrocytes, and microglia in the developing brain, and thus, resting microglia should be considered neuroglia, which are derived from neuroepithelia [10]. Another group insisted that microglia originate from the floor plate glial cells arising from the neuroepithelium, and thus, they concluded that microglia are of neuroectodermal origin [11]. Both these groups argued that microglia differentiate from neuroepithelia just as the other glial cell types do.

However, it is now clarified that microglia originate from yolk-sac-derived fetal macrophages that develop into the microglial precursors observed in early-stage murine embryos [12]. In rodents, these precursor cells exist mixed in the developing neuroectodermal cells as early as embryonic day 8 [13,14], when neovascularization is not yet completed. In the perinatal stages, two kinds of morphologically distinct microglia, ameboid microglia and ramified microglia, can be seen in the brain. The ameboid microglia have short or no processes growing from a relatively large cell body [15,16] and are observed particularly in the supraventricular corpus callosum and around ventricles. The ramified microglia include a few long prolongations extended from a small cell body and are seen in the cerebral cortex at the same stage. The ameboid microglia were thought to be derived from monocytes that entered from the blood vessel [15,17], and they have been shown to stay in place over the two weeks after birth. However, by three weeks postnatally, the ameboid microglia disappear and only the ramified microglia are seen throughout the CNS [18]. In the adult brain, microglia mostly take on the shape of ramified microglia and distribute to the CNS with regular spacing.

What is the relationship between ameboid microglia and ramified microglia? It is generally speculated that ameboid microglia change into ramified microglia during brain development [19,20,21]. On the other hand, some researchers have argued that ameboid microglia do not transform into ramified microglia, because a smooth transition from ameboid to ramified microglia is not observed during brain development. Thus, the transformation from ameboid microglia to ramified microglia has not been sufficiently explained.

As stated above, several points of controversy remain concerning the cell lineage and the transformation of microglia. These have engendered still more questions. If yolk-sac-derived fetal macrophages migrate to the early developmental brain, do they proliferate in the parenchyma? Are microglia always supplied from monocytes in the “bone-marrow-derived monocytes” theory? According to the “neuroectodermal cell origin” theory, do microglia themselves not proliferate? Contemporary biologists cannot satisfactorily answer these questions, presumably due to the insufficient information about microglial proliferation. Thus, whether microglia have an ability to proliferate and how the ability is regulated in the parenchyma have attracted the interest of microglial researchers. This review will address the phenomenon of microglial proliferation in response to neuron injury and the molecular mechanism by which the mitotic reaction is advanced in microglia.

## 2. The Proliferative Feature of Microglia

As described above, microglia in the normal adult CNS are present as ramified microglia accounting for a minor population among glial cells. Their cell density is kept very low in the normal CNS. Microglia are estimated to make up 5–12% of total cells in mouse brain [22]. Recent quantification revealed that microglia constitute approximately 7% of non-neuronal cells in the mammalian brain [23]. Courtney et al. estimated that microglia account for 4.5–5.5% of total cells in the mouse brain [24]. In addition, a meta-analysis of studies quantifying the cell number of glial cells in humans reported that human microglia are generally considered to make up around 10% of total glial cells [25].

Although microglia have characteristic biological features such as a competence for morphological change, migratory capacity, and phagocytic ability, they also have the notable ability to increase their number in response to various pathological events, including trauma [26,27], or diseases/disorders [28,29]. Such microglial features have caught the eye of researchers because the phenomenon of microglia increasing their cell numbers stands out in the CNS. In the human brain, increases in microglial cell number have been recognized in neurodegenerative diseases, including Alzheimer’s disease [30,31], Parkinson’s disease [32,33,34], multiple sclerosis [35,36], Huntington’s disease [37], stroke [38,39], ischemia [40], amyotrophic lateral sclerosis (ALS) [41,42], and prion diseases [43,44]. Increases in microglial cell number were also observed in lesioned sites including the injured spinal cord [45,46], axotomized hypoglossal nucleus [47,48], and the spinal cord undergoing rhizotomy or sciatic nerve transection [49] in rodents.

In this context, we should carefully consider the statement “microglia increase their number in response to various pathological events”. An increase in cell number in vivo usually occurs due to a migration from around tissue or cell proliferation at the site of insult or injury. Which event contributed to the increase in microglial cell number is an important question. In the usual lesion models, peripheral macrophages enter the injured parenchyma, and thus, it is hardly possible to distinguish the resident microglia-derived activated microglia from macrophages infiltrated from the periphery. Even if microglia-like cells are increased at the injured sites, we cannot say that they all came from resident microglia because they may contain the infiltrated macrophages in addition to activated microglia.

We have sometimes experienced such ambiguous results in our experiments, and inevitably the following questions arise. Did microglia alone proliferate at the site of trauma? Do macrophages also infiltrate into the injured place? How can you distinguish the resident microglia and infiltrated macrophages? These questions are related to the matter that the activated microglia and the peripheral macrophages express some common surface antigens and thereby they are not easily distinguished from each other. Thus, it remained to be determined whether the increase in microglial cells in lesioned sites was ascribable to the mitosis of microglia themselves or the infiltration of peripheral macrophages.

A peripheral nerve axotomy model was found to be appropriate for analyzing the proliferation of microglia. For example, axotomy of the rat facial nerve does not cause the infiltration of blood-derived cells in the ipsilateral facial nucleus [50,51,52], enabling us to examine the response of microglia alone in the parenchyma (Figure 1A). That being said, we need to take care when mice are used for this purpose. In the mouse facial nerve axotomy model, unlike in rats, significant amounts of blood-derived cells are known to infiltrate the parenchyma [53,54]. From this point of view, a rat model is suitable for analyzing the reaction of microglia alone in parenchyma. In fact, notable results have been achieved by using a rat facial nerve transection model.

## 3. Tools for Detecting Microglia

To analyze microglial proliferation in the parenchyma, we need markers that can distinguish microglia from other cells in the CNS. It was a weak silver carbonate staining method that del Rio-Hortega first used to distinguish microglia from other glial cells in the CNS. Although this staining method was reasonably effective when introduced, it is not used often as a routine method because it often yields unstable and unreliable results, meaning a very hit-and-miss staining. Moreover, alternative methods to reliably detect microglia have since been developed, and today we have a few practical methods involving enzyme histochemistry, lectin-binding, and antigen/antibody-binding.

For the enzyme histochemical reaction, thiamine pyrophosphatase (TPPase) and nucleoside diphosphatase (NDPase) [55,56] can be used. It is possible to stain reliably resting microglia (ramified microglia) by this reaction. A few lectins were found to have the ability to bind to microglia. However, we need to pay attention to the species-specificities. Isolectin B_4_ derived from *Griffonia simplicifolia* [57,58] and lectin from *Lycopersicum esculentum* [59] can stain rat microglia but not the microglia of other animals. Lectin from *Ricinus communis* binds to human and rat microglia [60,61]. These lectins are valid for detecting microglia while being careful concerning species.

Since microglia express a variety of surface antigens, antibodies against them were prepared and used for detecting/identifying microglia both in vivo and in vitro. F4/80, a monoclonal antibody specific for mouse microglia/macrophages, and 2.4G2, a monoclonal antibody specific for mouse Fc receptor recognized microglia in the mouse brain [62]. Antibody against complement receptor 3 (CR3)/CD11b stained not only activated microglia but also resting microglia [63]. Anti-CR3 antibody has proven effective for staining microglia in immunocytochemical or immunohistochemical studies. Some antibodies such as anti-major histocompatibility complex (MHC) class II antibody [64,65] can detect microglia in parenchyma, but the antibodies recognize only a portion of the microglia. From around 2000, anti-ionized Ca^2+^-binding adapter molecule 1 (Iba1) antibody [66,67] has been widely used. This antibody is available for immunohistochemical detection of not only activated (reactive) microglia but also ramified (resting) microglia (Figure 1B,C). It is also possible to recognize microglia in vitro by an immunocytochemical method and immunoblotting. By using these advantageous tools, it is now possible to detect and distinguish ramified and activated microglia using simple procedures and to collect information regarding their localization in the CNS.

## 4. Rat Facial Nerve Transection Model

What appearance do microglia take in the normal cerebral cortex? Anti-Iba1 antibody shows a typical shape of ramified microglia that bear some long processes. We can see such ramified morphology of microglia everywhere in the mature CNS (Figure 1B,C). Ramified microglia transform to activated microglia in response to various stimuli, including trauma, infection, and diseases. The activated microglia can be observed in the rat facial nerve transection model. The adult rat facial nerve was unilaterally transected, and at 5 days post-insult, both the contralateral and ipsilateral facial nuclei were immunohistochemically stained by using anti-Iba1 antibody. The avidin biotin complex (ABC) technique revealed that Iba1 is highly expressed in the axotomized facial nucleus (axotFN) (Figure 2A). In the control facial nucleus (contFN), only ramified microglia were seen. On the other hand, activated microglia with irregular shapes were increased in the axotFN (Figure 2B). The remarkable increase in microglia in the axotFN was recognized by immunohistochemistry using anti-CR3 (CD11b) antibody (Figure 2C). These antibodies were turned out to be reliable for staining ramified microglia and activated microglia in immunohistochemical studies.

Immunoblotting using Iba1 made it possible to analyze quantitatively the numbers of microglia. The analysis of the axotFN of adult rats (postnatal 8 week; p8W) whose facial nerves were unilaterally cut at 1, 3, 5, 7, and 14 days previously indicated that microglia increased their cell number from 3 days post-insult, and the cell number reached the maximum at 5–7 days post-insult (Figure 2D, p8W). On the other hand, the microglial cell number in the contFN did not change at any time. Since blood cells hardly infiltrate to parenchyma in rat injury models [50,51,52], the increase in microglial cells in the axotFN is ascribed primarily to the proliferation of microglia themselves.

As shown above, the microglial cell number in the axotFN of adult rats reached a maximum at 5–7 days post-insult (Figure 2D, p8W). Concerning the time course of the transition of the microglial cell number, one frequently asked question is: Does the microglial proliferation occur in newborn rats and aged rats in the same manner? In the facial nerve transection model using newborn rats, Graeber et al. reported that the microglial cell number increased in the axotFN with a peak at 5–7 days post-injury [68]. On postnatal day 2 (p2d)-rats, the increased cell number in the axotFN reached a peak at 5–7 days post-insult [69] (Figure 2C, p2d). How does this time course compare to the case of senescent (aged) animals? In postnatal-week-60 (p60W)-rats, the microglial cell number reached a maximum at 5–7 days following transection (Figure 2C, p60W). Hurley and Coleman compared microglial proliferation between 3-month-old and 21–25-month-old rats by immunohistochemical staining using anti-CR3 antibody and found that the increased levels of microglia in the axotFN were almost equivalent between the two [70]. In a comparison of the facial nerve axotomy models established using 3-, 15-, and 30-month-old rats, Conde and Streit verified that the microglia of the oldest rat (30 months) incorporated the highest level of ^3^H-thymidine in the axotFN [71], suggesting that aged microglia retain their ability to proliferate at a higher level. Their findings suggested that even aged rat microglia have the potential to proliferate rigorously.

## 5. Proliferating Factors for Microglia

The fact that microglia increased their cell number in the axotFN suggests that a specific growth factor(s) is induced and stimulates microglia to divide in the axotFN. What kinds of growth factor function in the microglial mitosis in the axotFN? Factors which contribute to the proliferation of microglia have emerged from in vitro experiments. To date, several kinds of molecules have been reported as proliferation factors for microglia.

The main group of microglial proliferation factors is the colony-stimulating factors (CSF), which include macrophage colony-stimulating factor (M-CSF; CSF1) [72,73,74], granulocyte macrophage colony-stimulating factor (GM-CSF) [75,76,77], G-CSF [78], and multipotential CSF (multi-CSF) [75,76,79].

Giulian et al. isolated microglial mitogens (MM), which promote microglial mitosis in the developing brain or injured brain [80]. The biochemical analysis revealed that MM1, a protein with a molecular mass of 50 kDa and pI of 6.8, displays GM-CSF-like activity, and MM2, which has a molecular mass of 22 kDa and pI of 5.2, is secreted from astrocytes but is different from interleukin-3 (IL-3). These results suggested that CSF-like factors serve as mitotic factors for ameboid microglia that are specifically present in the developing or injured brain.

In addition to CSFs and MMs, IL-4 [81] and IL-5 [82] were found to enhance microglial proliferation. Two inflammatory cytokines, i.e., tumor necrosis factor alpha (TNFα) and IL-1β, have an ability to promote the mitosis of microglia in the presence of astrocytes [76]. It is likely that these inflammatory cytokines (TNFα and IL-1β) influence microglial density under an inflammatory condition in the pathological state of the CNS. In this way, several factors, such as CSFs and cytokines, have been reported to promote the proliferation of microglia in in vitro experiments.

Apart from the results of in vitro experiments, attention has also been paid to which specific molecules contribute to microglial mitosis in the axotFN. Raivich et al. performed a series of breakthrough studies to identify microglial mitotic factors by analyzing the axotFN of osteopetrosis (op/op) mice [83], which cannot produce functional M-CSF due to a frameshift mutation [84] and showed that microglia hardly proliferated in the ipsilateral nucleus of these mice [85]. The results indicated that M-CSF is a crucial and a major mitotic factor of microglia in vivo. M-CSF [86,87] is a factor that stimulates the proliferation of bone-marrow-derived monocytes and macrophages, and also acts as a survival factor for them. M-CSF exists at low levels in the rat cerebral cortex and brainstem, and the low level of M-CSF is speculated to function in the survival, but not in mitosis, of ramified microglia in the same manner as monocytes/macrophages [86].

Immunoblotting results showed that the amounts of M-CSF were significantly increased in the axotFN at 3–5 days following axotomy [88] (Figure 3A). Immunohistochemical study revealed that anti-M-CSF antibody-positive cells were consistent with anti-CD11b antibody positive cells [88], indicating that M-CSF protein is produced in microglia in the axotFN. In the same report, GM-CSF, G-CSF and IL-3 were not significantly detected in the axotFN [88]. As for TNFα/IL-1β, which have the ability to promote the proliferation of microglia, they were not significantly detected in the axotFN [88], which agrees with a report in which the expression of TNFα/IL-1β mRNA was found to be minimal in the axotFN [89]. Accordingly, it was suggested that these factors with mitotic potential—namely, GM-CSF, G-CSF, IL-3, IL-1β, and TNFα—do not substantially serve as proliferation factors of microglia in the axotFN. Thus, a major mitotic factor of microglia in the axotFN was suggested to be M-CSF.

## 6. M-CSF: A Trigger of Microglial Proliferation

As described above, microglia were anticipated to proliferate in response to M-CSF in the axotFN. Thus, Yamamoto et al. analyzed what happens in the M-CSF-stimulated microglia using an in vitro system [88] in which microglia were prepared from a newborn rat brain-derived mother culture and used as neonatal microglia (neoMicroglia) [92].

The neoMicroglia were found to incorporate bromo-deoxy uridine (BrdU), indicating that they were mitotic cells. However, they entered a quiescent state when maintained with new medium. Such non-proliferative microglia were found to change to proliferative cells by stimulation with M-CSF. The M-CSF-stimulated microglia begin to increase their cell number after a 2-day delay (Figure 3B), and over the 3-day culture the cell number became approximately threefold the baseline value, depending on the concentration of M-CSF (0–40 ng/mL). The neoMicroglia increased their cell number in response to not only M-CSF but also GM-CSF and IL-3 (Figure 3C), as has been previously reported [76].

When the microglia of an adult animal received an interest from a viewpoint of cellular characteristics, Rieske et al. succeeded in culturing adult axotFN-derived microglia [93]. This study first established an explant culture to prepare the adult microglia. Owing to the method, axotFN-derived microglia (axotMicroglia) can be obtained with high purity [91]. These axotMicroglia were intrinsically proliferative, and thus, they increase in number in culture without addition of any growth factors (Figure 3D). Over the 10-day culture, their number increased sevenfold or more (Figure 3E). The cultured microglia vigorously incorporated BrdU in their nuclei, suggesting that they undergo mitosis. Analysis of axotMicroglia in vitro clarified that the M-CSF protein was present in both the cells and the conditioned medium. The receptor of M-CSF, cFms, was also found to be expressed in the microglia [91]. The axotMicroglia appeared to autonomously proliferate, suggesting that they proliferate in such a manner that they produce/secrete M-CSF and respond to it—that is, in an autocrine fashion.

## 7. Cell-Cycle Associated Proteins in Proliferating Microglia

We will next consider the molecular mechanism by which microglia proliferate in the axotFN. Since mitosis progresses according to the functions of growth factors/receptors, cell-cycle associated proteins, signal-mediating molecules, and transcription factors, these molecules were analyzed in microglia in the axotFN.

As noted above, a major growth factor of microglia M-CSF [86,87] was transiently increased in the axotFN with a peak at 3 days post-insult [88], which is a little earlier than the time of the mitotic peak. Immunohistochemistry revealed that M-CSF was expressed in microglia. The receptor of M-CSF cFms (CSF1R) was markedly enhanced in the axotFN from 3 days post-insult [88] (Figure 4A). cFms was originally discovered as a proto-oncogene [94], and then later identified as a CSF-1 receptor (M-CSF receptor) [95]. cFms is a tyrosine kinase receptor, and in response to M-CSF or IL-34 [96] the tyrosine residue of cFms in the cytoplasm is phosphorylated, to which SH2-domain-expressing molecule binds and transmits a signal to downstream. The increase in M-CSF receptors in the axotFN was first verified by Raivich et al. [27,97]. They clarified immunohistochemically that cFms is expressed in microglia. The levels of cFms in microglia in the contFN are very low, but the levels became high in the axotFN [88].

To examine whether the cell-cycle is promoted in proliferating microglia, the levels of proliferating cell nuclear antigen (PCNA) were analyzed in the axotFN. PCNA is a nuclear protein that acts as processivity factor for DNA polymerase δ in eukaryotic cells and is mainly induced in the S phase [98,99]. Accordingly, PCNA is used as an S-phase marker. The levels of PCNA were promoted in the axotFN from 3–7 days post-insult (Figure 4A), suggesting that DNA replication occurred at 3–7 days post-insult. PCNA was found to be expressed in microglia (Figure 4B), but not astrocytes in the axotFN [88]. Astrocytes do not proliferate in the adult axotFN, although they do proliferate together with microglia in the neonatal axotFN [68].

Cyclin forms the M-phase-promoting factor (MPF) with cyclin-dependent kinase (Cdk) as cyclin/Cdk [100,101] and progresses the cell cycle, and when the stage ends, the cyclins are degraded. Cyclin A/Cdk2 [102,103] functions in the S phase and G_2_/M transition, and cyclin D/Cdk4 or cyclin D/Cdk6 generally promotes the transition from G_1_ to S phase [104]. Sherr reported that M-CSF stimulation elevated the levels of cyclin D in macrophages [105], inferring that cyclin D is functioning in microglial proliferation in the axotFN. In the axotFN, cyclin A and cyclin D were detected, and they were found to transiently increase at 3–7 days after nerve injury [90] (Figure 4A), in agreement with the time of upregulation of PCNA. An immunohistochemical study confirmed that these cyclins were enhanced in microglial cells in the axotFN. Whether the cyclins play a role in the microglial proliferation was investigated in M-CSF-stimulated microglia by using the Cdk2 inhibitor purvalanol A (PA) [106], which has been shown to inhibit the activity of MPFs including cyclin A/Cdk2, cyclin B/Cdk2, and cyclin E/Cdk2. The results indicated that the microglial proliferation was significantly suppressed by pretreatment with PA, suggesting that the induced cyclin A/Cdk2 activity serves in the cell division of microglia in the axotFN.

There was an issue concerning the transient proliferation of microglia. The induction of M-CSF/cFms and cyclin A/D/E in microglia in the axotFN allowed us to predict that microglia continue to proliferate for a long time following nerve injury. In fact, however, the cell number of microglia reached a maximum at 5–7 days post-injury and then decreased—that is, it showed transient proliferation—as shown above [90]. This raises the question: Why was the microglial cell number not maintained at high levels for a much longer period after injury? This question predicted the existence of an inhibitory mechanism by which the progression of the cell cycle is impeded. As the results of an investigation, an intrinsic inhibitor of Cdk, p21, was highlighted in the axotFN. The Cip/Kip family protein, p21, is a Cdk inhibitor that acts on the Cdk/cyclin complex [107,108]. p21 was detected in the axotFN, and it was rapidly induced at 5 days post-insult [90]. This result strongly suggests that the proliferation of microglia was inhibited by the function of p21, so that the number of microglial cells was not increased too dramatically.

## 8. Signaling Molecules Serving in Proliferating Microglia In Vivo

The fact that some cell-cycle-related molecules are upregulated in proliferating microglia in the axotFN suggested that a specific signaling cascade is involved in the upregulations. In the analysis of mitogen-activated protein kinases (MAPKs) in vivo, phosphorylated p38 (p-p38) was found to be remarkably increased in the axotFN at from 1 d to 14 days post-insult [109] (Figure 4C). The levels of p38 also increased from 3 to 14 days post-insult. An immunohistochemical study indicated that p-p38 was exclusively expressed in the axotFN [109] (Figure 4D). These p-p38-expressing cells were consistent with those of CD11b-expressing cells, demonstrating that p-p38 was expressed in growing microglia. It appeared likely that p-p38 functioned in the mitotic reaction of microglia.

Furthermore, the phosphorylated cAMP-responsive element binding protein (p-CREB) was extensively found in the nuclei of many small cells in the axotFN [109] (Figure 4E). Though p-CREB-positive cells were not directly identified due to the inconvenient combination of antibodies, CREB-positive small cells were certified as microglia [109]. Thus, CREB was predicted to be highly activated as p-CREB in microglia and the p-CREB was shown to engage in the mitotic reaction of microglia in the axotFN. The transcription factor CREB is generally involved in a variety of cellular processes, including differentiation, survival, and proliferation [110,111]. In the case of microglia in the axotFN, p-CREB/CREB was suggested to function in the transcription of mitosis-related genes downstream of a MAPK, p38 (described below).

Activating transcription factor 2 (ATF2) is a transcription factor serving in various cellular responses such as anti-apoptosis reaction, cell growth, and DNA damage [112,113]. ATF2 has been shown to be phosphorylated in lipopolysaccharide (LPS)-stimulated microglia both in vitro and in vivo [114], suggesting that the activated ATF2 participates in microglial activation. In fact, a greater number of p-ATF2-expressing small cells were observed in the axotFN (Figure 4F). Anti-ATF2 antibody-positive small cells were also elevated in the axotFN. The small cells were identified as microglia by the dual staining with anti-ATF2 antibody and anti-CD11b antibody [109]. Thus, the increase in p-ATF2/ATF2 in the microglia in the axotFN was suggested to be associated with the mitotic reaction.

## 9. Signaling Molecules Serving in Microglial Proliferation In Vitro

In the axotFN, M-CSF, cFms, PCNA, cyclin A/D, and p21 were identified as mitosis-related molecules, and p-p38/p38, p-CREB/CREB, and p-ATF2/ATF2 were found to be mitosis-related signaling molecules in the proliferating microglia. What link is there between the mitosis-related molecules and the signaling molecules in proliferating microglia? The relationship between them became clear in an in vitro experiment using neoMicroglia.

Non-stimulated neoMicroglia express only a limited amount of cFms, PCNA, and cyclin A/D, and they do not increase in cell number in the absence of growth factors. However, when stimulated with M-CSF, the microglia begin to change into a mitotic state from a quiescent state. The first step in this process is the binding of M-CSF to the specific receptor cFms, by which cFms is rapidly phosphorylated. Tyr-phosphorylation of cFms was observed within 1 min after M-CSF-stimulation. Subsequently, microglia enhanced the amounts of cFms, PCNA, and cyclin A/D. After 24 h, the levels of cFms, PCNA, cyclin A, and cyclin D were increased approx. 10-fold, 11-fold, 6.5-fold, and 5-fold, respectively [90]. If a cFms receptor tyrosine kinase inhibitor GW2580 (GW) is added to the M-CSF-dependent microglial proliferation system, microglia do not increase the cell number, and simultaneously do not enhance the levels of cFms, PCNA and cyclins [90]. These results indicated that the M-CSF/cFms reaction is a trigger for microglial proliferation.

In the M-CSF-dependent microglial proliferation system, three MAPKs, extracellular signal-regulated kinase (ERK), c-Jun N-terminal kinase (JNK), and p38, were all phosphorylated within 5 min following M-CSF stimulation [90]. The phosphorylation of JNK and p38 in M-CSF-stimulated microglia was significantly inhibited by treatment with GW, suggesting that M-CSF, cFms, and JNK/p38 are linked as M-CSF–cFms–JNK/p38 cascades (Figure 5). The role of each MAPK on the induction of proliferation-related molecules was investigated by using specific inhibitors. A specific inhibitor of MEK 1/2 (an upstream kinase of ERK), PD98059 [115], did not affect the levels of cFms, PCNA, or cyclins in M-CSF-stimulated microglia, suggesting that p-ERK/ERK does not play a role in the induction of mitosis-related molecules. In contrast, administration of the JNK inhibitor SP600125 [116] to M-CSF-stimulated microglia suppressed the levels of PCNA and cyclin A/D but not the levels of cFms [90]. These results suggested that JNK is connected to the signaling cascade leading to the induction of PCNA and cyclin A/D (Figure 5). Treatment of microglia with a p38 inhibitor, SB203580 [117], reduced the level of cFms, but not the levels of PCNA and cyclin A/D, in M-CSF-stimulated microglia [90], suggesting that p38 is involved in the induction of cFms (Figure 5). Thus, among MAPKs, JNK was suggested to play a role in the induction of cell-cycle related proteins, including PCNA and cyclin A/D, whereas p38 was suggested to serve in the induction of cFms.

As described above, immunohistochemical analysis showed that microglia expressing p-CREB and p-ATF2 were significantly enhanced in the axotFN. An in vitro experiment also documented that the levels of p-CREB and p-ATF2 were enhanced in M-CSF-stimulated microglia (partially unpublished). In addition, mitogen- and stress-activated kinase-1 (MSK1) was recognized to be phosphorylated in M-CSF-stimulated microglia in vitro [109]. Since MSK1 is phosphorylated by p38 [118,119] and the activated MSK1 (p-MSK1) in turn activates CREB [120], the p38–MSK1–CREB signaling cascade is assumed to serve in microglial proliferation. ATF2 was also phosphorylated in M-CSF-stimulated microglia in vitro, and a link between p38 and ATF2 was reported in microglia in an LPS-induced neuroinflammation model [114], suggesting the existence of a p38–ATF2 pathway in M-CSF-dependent microglial proliferation. Thus, it is plausible that p38–MSK1-CREB and/or p38–ATF2 cascades are involved in microglial proliferation through the induction of cFms [90].

Therefore, it is possible to depict the signaling pathway through which microglia proliferate in response to M-CSF (Figure 5). In the figure, MAP kinase kinase (MKK)4/7 and MKK3/6 were added upstream of JNK and p38, respectively, since these kinases are found to be phosphorylated in M-CSF-stimulated microglia (unpublished data). In Figure 5, we can see the signaling molecules and signaling cascades functioning in the microglial mitosis occurring in the axotFN.

## 10. Significance of Proliferating Microglia

As shown in Figure 2E, microglia in the axotFN proliferate and increase the in number, simultaneously an interesting image displaying round localization of microglia. The image shows that microglia enclose the injured motoneuronal cell bodies These distributions of microglia are called satellites. The significance of “microglial satelliting” has remained obscure. The characteristic location of microglia allowed us to investigate the properties of microglia to produce biologically active substances including survival and/or trophic factors for neurons [121,122].

Early analysis of growth factors in the axotFN demonstrated that transforming growth factor beta 1 (TGFβ1) was enhanced in activated microglia [123]. The role of enhanced levels of TGFβ1 was analyzed in TGFβ1-deficient mice [124], and the TGFβ1 was suggested to serve in neuroprotection as well as microglial proliferation. Several reports have described that the proliferating microglia in the axotFN supply neurotrophic factors to injured motoneuron cell bodies at point-blank range. In the axotFN, mRNA levels of brain-derived neurotrophic factor (BDNF) receptor (TrkB) are increased in injured motoneurons from 2 days to 3 weeks after axotomy [125]. The mRNA of glial-cell-line-derived neurotrophic factor (GDNF) receptor components (GFR-a1 and c-ret) was increased at 1–3 days post-insult [126]. The levels of leukemia inhibitory factor receptor beta (LIF-Rβ) mRNA/protein were increased in injured motoneurons after axotomy [127]. These results do not directly mean that ligands such as BDNF, GDNF, and LIF are produced/secreted from microglia, but they do allow us to predict that injured motoneurons attempt to trap as many ligands released from satelliting microglia as possible. It is most likely that the microglia rescue motoneurons from degeneration by supplying neurotrophic factors from close-range because microglia were verified to have the capacity to produce a variety of neurotrophic factors in an in vitro system [128]. Microglia (neoMicroglia) have been shown to produce neurotrophins, including nerve growth factor (NGF); BDNF; neurotrophin-3 (NT-3) and NT-4/5 [129,130]; TGFβ superfamily proteins, including TGFβ1 [131] and GDNF [132]; and ciliary neurotrophic factor (CNTF)/LIF/IL-6 family proteins, including CNTF [133], LIF, and IL-6 [134]. In addition, hepatocyte growth factor (HGF) [135], insulin-like growth factor-1 (IGF-1) [136], and IL-3 [137] were reported to be produced in microglia. The ability of microglia to produce/secrete neurotrophic factors was suggested to be promoted by neurons. It is reported that the neuronal conditioned medium (NCM) enhanced the secretion from microglia of neurotrophic factors, including NGF, NT4/5, TGFβ1, GDNF, fibroblast growth factor 2 (FGF2), and IL-3 [138], supporting the notion that the satelliting microglia supply more neurotrophic factors through close interaction with motoneurons.

Proliferating microglia taken from axotFN were demonstrated to exert neurosurvival effects on rat-brain-derived neurons [91]. Conditioned medium (CM) of axotFN-derived microglia protected against the neuronal cell death that occurred to some extent in the culture. The survival effect was comparable to that of neoMicroglia-derived CM. The analysis of axotFNmicroglia and their CM revealed that they produce neurotrophins (BDNF and NT-4/5) and GDNF. In addition, neoMicroglia stimulated with M-CSF were found to enhance the amounts of GDNF in vitro. These results suggest that microglia proliferating in the axotFN contribute to the supply of neurotrophic factors to injured motoneurons in a paracrine fashion.

In addition to neurotrophic features, it has become increasingly clear that the microglia work to remove harmful materials/molecules that damage motoneurons in the milieu of the facial nucleus. Viewed from the perspective of excitatory neuronal cell death, microglial satelliting around injured motoneurons was suggested to prevent excitatory neuronal cell death. If Glu is highly released from the presynaptic membrane and remains for an extended period in the synapse, glutamate receptors (GluRs) in postsynaptic neurons would be extraordinarily stimulated, and the neurons would undergo cell death [139,140]. To prevent this neuronal cell death, two glial type-Glu transporters, glutamate aspartate transporter (GLAST; EAAT-1) and glutamate transporter-1 (GLT-1; EAAT-2) [141,142] are prepared in vivo. Satellite microglia in the axotFN have been shown to express high levels of GLT-1 [143], suggesting that the microglia eliminate Glu locally by upregulating GLT-1. The ability of microglia to uptake Glu by GLT-1 was demonstrated in in vitro experiments [144]. In another study, a co-culture experiment revealed that microglia upregulated the level of GLT-1 and enhanced the uptake of Glu by GLT-1 in the presence of neurons [145]. In that study, a neuron-derived soluble molecule(s) was conjectured to provoke microglia to promote the elimination of Glu by increasing GLT-1 levels. AxotFN-derived microglia also showed the ability to uptake extracellular Glu [146]. How is the Glu taken into microglia metabolized? Biochemical analysis using nuclear magnetic resonance (NMR) indicated that ^13^Glu is changed to ^13^Gln in microglia by the function of Gln synthetase [146], suggesting that at least a part of Glu taken into microglia is metabolized to Gln. In short, we can say that GLT-1-expressing microglia in the axotFN scavenge dangerous Glu and metabolize it to safe Gln, eventually protecting injured motoneurons from abnormal Glu-induced cell death.

Oxidative stress is generally implicated in the pathogenesis of degenerative diseases [147,148], including Alzheimer’s disease, Parkinson’s disease, and ALS [149]. The axotomized facial nucleus is no exception. The injured motoneurons in the facial nucleus undergo oxidative stress triggered by the injury [150]. In this case, microglia are a primary candidate for protecting motoneurons from the oxidative stress through cell–cell interactions [151]. The ability of microglia to prevent oxidative stress of neurons was documented in an in vitro experiment [152]. In another report, microglia were presumed to act as anti-oxidant and anti-inflammatory cells by expressing natural-resistance-associated macrophage protein 1 (Nramp1) [153] and heme oxygenase 1 (HO1) [154]. Generally, microglia are believed to accumulate high levels of glutathione, superoxide dismutase, catalase, and glutathione peroxidase/reductase for the protection against oxidative damage [155,156]. These properties would contribute to the protection of not only microglia themselves but also around neurons from oxidative stress. As a whole, microglia proliferating in the axotFN are believed to be neuroprotective cells rather than deleterious cells. These properties are easily imaginable based on the fact that injured motoneurons do not undergo cell death in the axotFN of adult rats.

## 11. Prospects

In this article, we examined the phenomenon that microglia undergo mitosis in the axotFN and analyzed it from the viewpoints of mitosis-related molecules and signaling molecules. The above-described series of studies provide a general picture of microglial proliferation as an outstanding event in the injured CNS.

However, a few issues remain to be solved. One is the question of the stimulus that induces microglia to produce/secrete M-CSF in the axotFN. M-CSF was found to be temporarily increased in the axotFN, and the factor was expressed in activated/proliferating microglia. Analysis of axotMicroglia revealed that some amounts of M-CSF are actually present in the microglia. These results suggest that a certain signal emitted from injured motoneurons stimulates microglia to produce/release M-CSF. Unfortunately, in vivo experiments have yielded no information about a signal released from injured motoneurons. Moreover, no M-CSF-inducing factor has yet been detected in in vitro experiments examining various biologically active molecules for their ability to induce M-CSF in microglia. Thus, this point remains to be settled.

The second issue is the difficulty of designing an experiment to clarify signaling cascade in vivo. Since the CNS comprises heterogenous cell types, including neurons and glial cells, it is obviously impossible to administer a specific inhibitor and/or activator to a specific cell type alone in vivo. For example, even if we aim to know the effects of an individual inhibitor on microglia in the axotFN, it would not be possible to conduct such an experiment. For that reason, there is no choice but to obtain a limited information primarily collected from general experiments including immunohistochemistry and an in vitro culture system. Such steady research will be needed to succeed in future investigations on the axotFN. We anticipate obtaining more detailed knowledge regarding the molecular mechanism of microglial proliferation someday.

## 12. Conclusions

A characteristic feature of microglia is their ability to proliferate in response to neuronal insult. The proliferation of microglia was analyzed in the axotFN, in which blood–brain barrier is kept healthy. Microglia begin to increase in number after nerve injury, with the cell number reaching a maximum at 5–7 days post-insult. The analysis of the microglial proliferation in the axotFN revealed that microglia exposed to an injury stimulus from injured motoneurons start to increase the level of M-CSF, which in turn stimulates cFms. Downstream of cFms, JNK and p38 are activated and serve in the induction of PCNA/cyclins and cFms, respectively. JNK was assumed to function in the signaling cascade leading to the induction of cell-cycle-related proteins, while p38 was predicted to be involved in the induction of cFms through a signaling cascade of p38-MSK1-CREB and/or p38-ATF2. In this way, the mechanism of microglial proliferation became clearer in the facial nerve injury model. Microglia proliferating at around injured motoneurons in the axotFN are suggested to contribute to the neuroprotection by supplying neurotrophic factors and/or eliminating excess amounts of Glu and reactive oxygen species.

## Figures and Tables

**Figure 1 biology-12-01121-f001:**
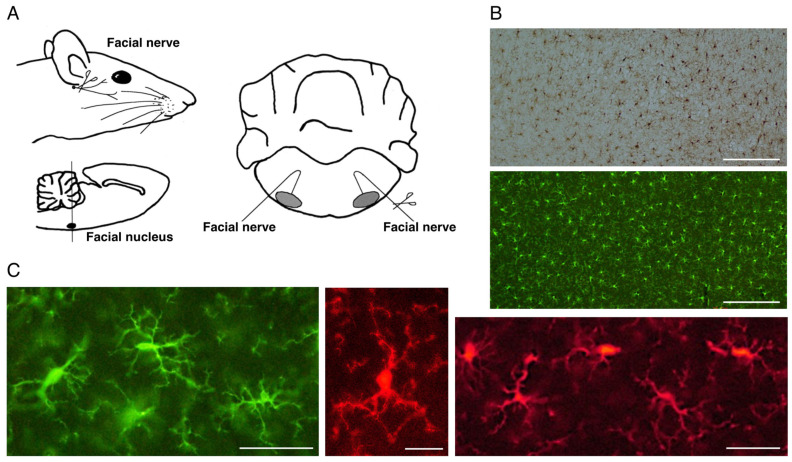
The rat motoneuron injury model and ramified microglia in the normal state. (**A**). Schema of rat facial nerve axotomy. Motoneurons in the facial nucleus of the brainstem extend their nerves to facial expression muscles. In the axotomy model, the facial nerve was unilaterally cut at the stylomastoid foramen, and subsequently both facial nuclei were analyzed. (**B**). Distribution of ramified microglia in the normal adult rat cerebral cortex. Ramified microglia were stained by the avidin biotin complex (ABC) method (upper panel) and fluorescence method (lower panel). Scale bar = 200 μm. (**C**). Morphology of ramified microglia. Typical morphology of ramified microglia is shown. Scale bar = 50 μm (left and right) or 20 μm (center).

**Figure 2 biology-12-01121-f002:**
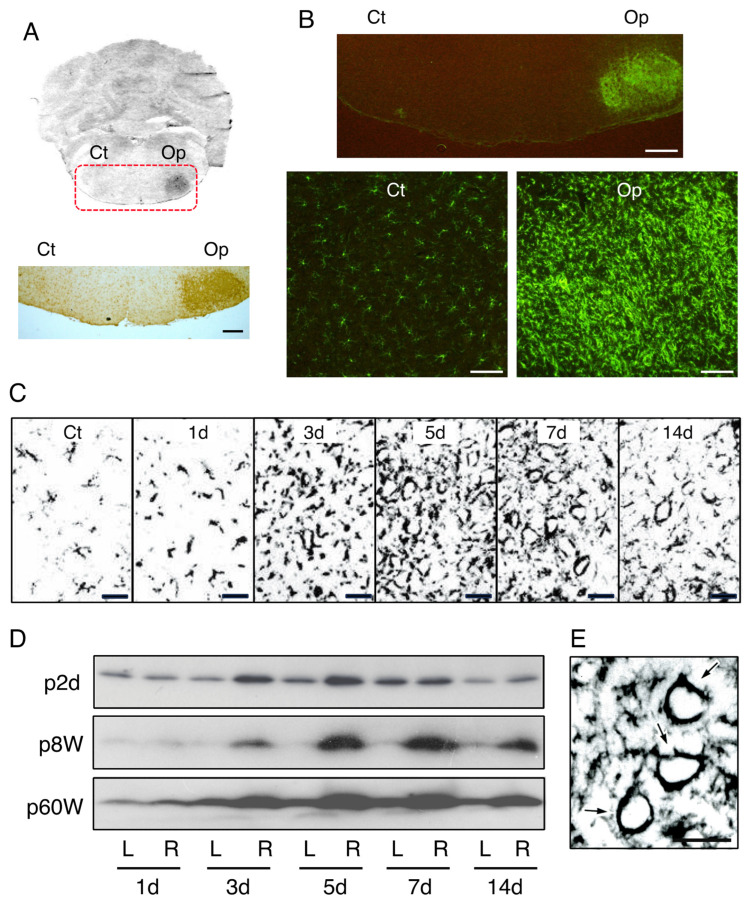
Increase in microglial cell number in the axotomized facial nucleus. (**A**). Amounts of microglia in contFN and axotFN. The right facial nerve was transected and 5 days later the brainstem sections were stained with anti-Iba1 antibody according to the ABC method. Ct, contFN; Op, axotFN. Scale bar = 400 μm. (**B**). Increase in microglia in the axotFN. The brainstem sections prepared as described above (**A**) were stained with anti-Iba1 antibody by the fluorescence method. Scale bar = 400 μm. High-power photos are shown in the lower panels. Scale bar = 100 μm. (**C**). Immunohistochemical analysis of the microglial cell number across time. The axotFN was stained with anti-CR3 antibody at 1, 3, 5, 7, and 14 days post-insult. Ct shows the contFN obtained at 5 days post-insult. Scale bar = 50 μm. (**D**). Immunoblot analysis of Iba1 in rats of different ages. The right facial nerve of a p2d-rat (p2d), p8W-rat (p8W), and p60W-rat (p60W) was transected, and the contFN (L) and axotFN (R) of each rat were immunoblotted for Iba1 across time. (**E**). Circular localization of microglia. Iba1-expressing microglia around injured motoneuron cell bodies at 7 days post-insult are shown (arrows). Scale bar = 50 μm.

**Figure 3 biology-12-01121-f003:**
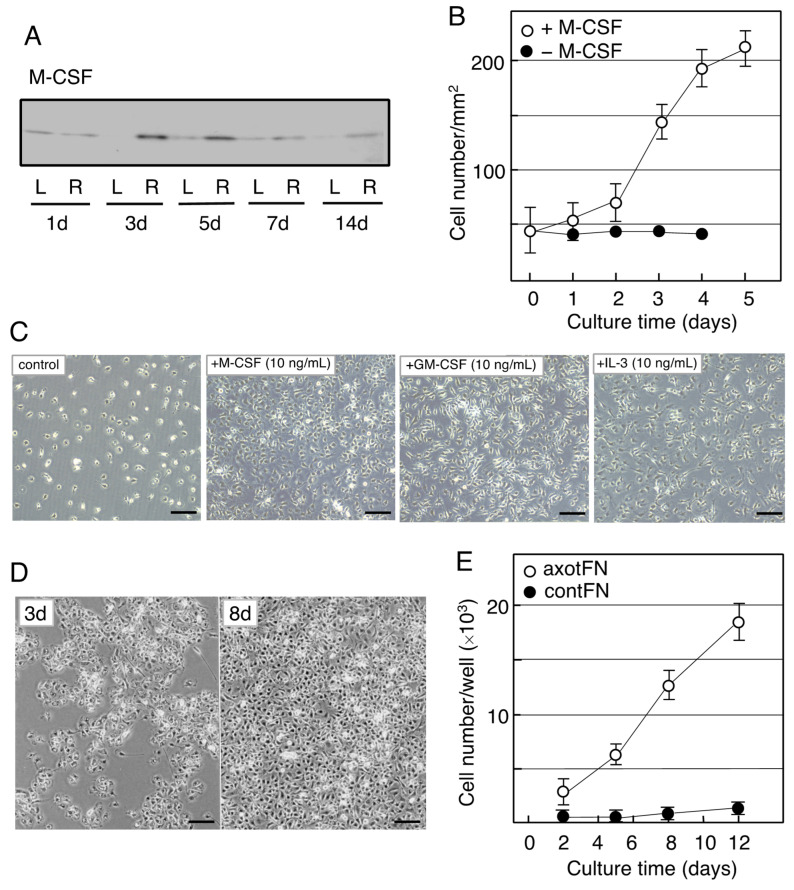
Induction of M-CSF in the axotomized facial nucleus and the effects of M-CSF on microglia. (**A**). Changes in M-CSF levels in the axotFN. The right facial nerve of adult rats was transected, and the contFN (L) and axotFN (R) of each rat were analyzed for M-CSF in immunoblotting at 1, 3, 5, 7, and 14 days post-insult. (**B**). Effects of M-CSF on microglial proliferation. M-CSF (20 ng/mL) was added to the microglial culture (2 × 10^4^/well) and the cell number within a 1 mm^2^ area (○) was counted [90]. The cell number in the absence of M-CSF is shown as (●). The results are expressed as means ± SDs of three separate experiments. (**C**). Effects of M-CSF, GM-CSF, and IL-3 (multi-CSF). M-CSF (10 ng/mL), GM-CSF (10 ng/mL), and IL-3 (10 ng/mL) were added to microglial culture and maintained for 3 days. Scale bar = 50 μm. (**D**). Increase in the cell number of axotFN-microglia. AxotFN recovered at 3 days post-insult were subjected to explant culture [91]. The microglia attached to the dish were photographed at 3 and 8 days after-culture. Scale bar = 50 μm. (**E**). Proliferation of axotFN-microglia. The axotFN transected 3 days previously (axotFN; ○) and contFN (contFN; ●) were subjected to explant culture [91]. The emerged cells were counted at 2, 5, 8, and 12 days following culture. The results are expressed as means ± SDs of three separate experiments.

**Figure 4 biology-12-01121-f004:**
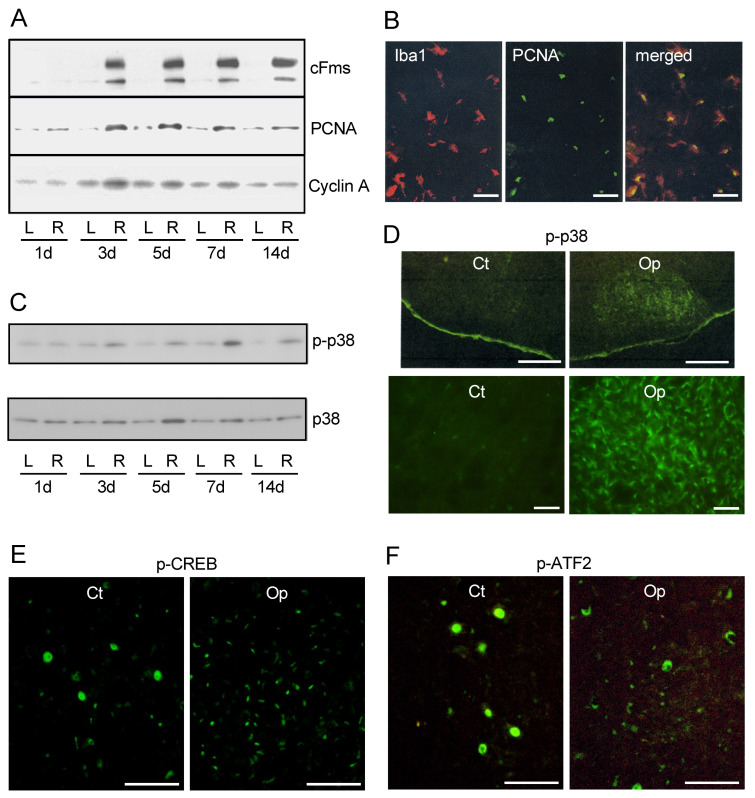
Cell cycle-related proteins and signaling molecules in the axotomized facial nucleus. (**A**). Changes in cell-cycle related proteins. Levels of cFms, PCNA, and cyclin A were analyzed in the contFN (L) and axotFN (R) recovered at 1, 3, 5, 7, and 14 days post-insult. (**B**). Expression of PCNA in proliferating microglia. The axotFN taken at 3 days post-insult was immunohistochemically dual stained with anti-Iba1(red) and anti-PCNA (green) antibodies. The merged photo is shown at right. Scale bar = 50 μm. (**C**). Transition of p-p38/p38 levels in the axotFN. The contFN (L) and axotFN (R) recovered at 1, 3, 5, 7, and 14 days post-insult were analyzed for p-p38 and p38 by immunoblotting. (**D**). Immunohistochemistry for p-p38 in the axotFN. Brainstem sections recovered at 5 days post-insult were stained with anti-p-p38 antibody. The contFN (Ct) and axotFN (Op) are shown in low (scale bar = 400 μm) and high magnification (Scale bar = 100 μm). (**E**,**F**). Immunohistochemistry for p-CREB and p-ATF2 in the axotFN. Brainstem sections recovered at 5 days post-insult were stained with anti-p-CREB antibody (**E**) and anti-p-ATF2 antibody (**F**). Note that the larger area of staining are nuclei of motoneurons, and the smaller stained points are nuclei of microglia. Scale bar = 100 μm.

**Figure 5 biology-12-01121-f005:**
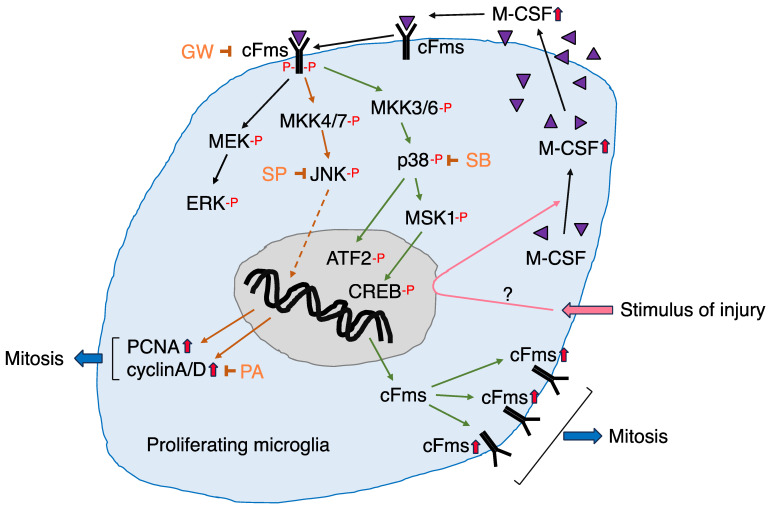
Schematic diagram of the signaling cascade in microglia proliferating in the axotomized facial nucleus. All abbreviations are as described in the text. 
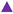
: M-CSF.

## Data Availability

Data are available in a publicly accessible repository.

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
