# Peer review of "Mechanisms of Microglia Proliferation in a Rat Model of Facial Nerve Anatomy"

_biology, 2023, doi:10.3390/biology12081121_

Round 1
Reviewer 1 Report
I have enjoyed reading this review article. Very nice, clear and organized.
Just a few comments for improvement:
*The simple summary is a replica to the abstract, if it is intended to be a lay summary, less technicalities should be used.
**In the introduction section is about the origins of the microglia, but lacks of a last paragraph describing what is going to be the topic of the review, and an intro/guide for the reader of what are going to be the next paragraphs. Also a short paragraph in the abstract would be helpful.
***The blots on page 6 figure 2D (p2d) appear with very poor quality and pixelated, please improve the equality of them.
****In the section of Future prospects, when talking about the fact that the use of specific inhibitors of microglia is impossible, have you considered strategies such as PLX5622, specific genetic animal models or the use of AAV for specific cell manipulation?
Author Response
Reviewer 1
I have enjoyed reading this review article. Very nice, clear and organized.
Just a few comments for improvement:
[comment 1]
*The simple summary is a replica to the abstract, if it is intended to be a lay summary, less technicalities should be used.
[response 1]
We appreciate the reviewer's comment. We have rewritten the simple summary (lines 8-18, page 1 in the revised manuscript).
[comment 2]
**In the introduction section is about the origins of the microglia, but lacks of a last paragraph describing what is going to be the topic of the review, and an intro/guide for the reader of what are going to be the next paragraphs. Also a short paragraph in the abstract would be helpful.
[response 2]
We see the reviewer's point. We added a paragraph describing the main topic of the review (lines 93-104, page 3 in the revised manuscript). We then deleted the former concluding paragraph of the “Introduction”, because it was no longer appropriate.
In addition, we added a sentence in the abstract (lines 19-22, page 1 in the revised manuscript).
[comment 3]
***The blots on page 6 figure 2D (p2d) appear with very poor quality and pixelated, please improve the equality of them.
[response 3]
We apologize for the low-quality blot image. We replaced it with an image of better quality (modified Fig. 2D; page 7).
[comment 4]
****In the section of Future prospects, when talking about the fact that the use of specific inhibitors of microglia is impossible, have you considered strategies such as PLX5622, specific genetic animal models or the use of AAV for specific cell manipulation?
[response 4]
Thank you for your suggestions on our experiment. Since PLX5622 was first reported, we have been interested in the microglia-inhibitory effects of PLX5622 in the CNS. Our interest has focused on whether the axotomized facial motoneurons can survive and regenerate in the absence of microglia. From the viewpoint of the neurotrophic effects of microglia, PLX5622 may indeed be a useful tool. However, we feel that it would not be as effective for our upcoming research, since we will be focused on analyzing the signaling cascades in the proliferating microglia.
The adeno-associated virus (AAV) method is known to be useful for expressing a specific gene in a targeted cell. Thus, if we want to express a proteinaceous p38 inhibitor in microglia in vivo, the AAV would be useful, because p38 activity in microglia is inhibited. Unfortunately, to our knowledge there is no intrinsic proteinaceous inhibitor for inhibiting p38.
In order to continue our study of the signaling cascades in proliferating microglia, we plan to determine the activation of signaling molecules in microglia by modulating the activity of signaling molecules in injured motoneurons. In a preliminary experiment, we found that the accumulation of neurotrophic factors or neurotoxins at the injured site change the levels of microglial proliferation. Thus, by monitoring the levels of activated signaling molecules in the motoneuron injury system, we could clarify some specific signaling molecules closely associated with the microglial mitosis.
Reviewer 2 Report
This review article summarizes the advances in biological research of microglia proliferation in response to facial nerve anatomy in rats and the potential cellular and molecular mechanisms of microglia activation in the facial nucleus. Notably, in the section of prospects the authors point out 1) the lack of stimulus that induces microglia to produce/secrete M-CSF for their mitosis in the axotomized facial nucleus, and 2) the limitation of using results from in vivo and in vitro experiments to solidate the molecular mechanisms by which autocrine M-CSF up-regulates microglia mitosis.
One major comment:
The cellular and molecular mechanisms of microglia proliferation may differ among species, life stages, locations in the central nervous system, and conditions of diseases/injuries. This article mainly reviews the phenomenon of microglia proliferation in response to facial nerve anatomy in rats and the potential mechanisms of microglia proliferation in the facial nucleus of rats. Therefore, it will be better that the title of this review article specifically reflects its primary contents - mechanisms of microglia proliferation in a rat model of facial nerve anatomy.
Minor comments:
1) The subtitle “Future prospects” should be “Prospects”.
2) In line 81, the author stated with citations of references: In the adult brain, microglia take on the shape of "ramified microglia"[15,17]. In fact, in the early postnatal ages microglia in rodents display ramified shapes.
Author Response
Reviewer 2
This review article summarizes the advances in biological research of microglia proliferation in response to facial nerve anatomy in rats and the potential cellular and molecular mechanisms of microglia activation in the facial nucleus. Notably, in the section of prospects the authors point out 1) the lack of stimulus that induces microglia to produce/secrete M-CSF for their mitosis in the axotomized facial nucleus, and 2) the limitation of using results from in vivo and in vitro experiments to solidate the molecular mechanisms by which autocrine M-CSF up-regulates microglia mitosis.
One major comment:
[comment 1]
The cellular and molecular mechanisms of microglia proliferation may differ among species, life stages, locations in the central nervous system, and conditions of diseases/injuries. This article mainly reviews the phenomenon of microglia proliferation in response to facial nerve anatomy in rats and the potential mechanisms of microglia proliferation in the facial nucleus of rats. Therefore, it will be better that the title of this review article specifically reflects its primary contents - mechanisms of microglia proliferation in a rat model of facial nerve anatomy.
[response 1]
This is a reasonable suggestion. Per the reviewer's recommendation, we changed the title to "Mechanisms of microglia proliferation in a rat model of facial nerve anatomy" (lines 2-3, page 1 in the revised manuscript).
Minor comments:
[comment 1]
1) The subtitle “Future prospects” should be “Prospects”.
[response 1]
We changed “Future prospects” to “Prospects” (line 581, page 17 in the revised manuscript).
[comment 2]
2) In line 81, the author stated with citations of references: In the adult brain, microglia take on the shape of "ramified microglia"[15,17]. In fact, in the early postnatal ages microglia in rodents display ramified shapes.
[response 2]
We are sorry that we did not correctly explain this point. Certainly, ramified microglia are present in the early postnatal stage in addition to ameboid microglia. Thus, we rewrote and modified the content (lines 75-85, page 2 in the revised manuscript).